# Withania somnifera (L.) Dunal whole-plant extracts exhibited anti-sporotrichotic effects by destabilizing peripheral integrity of Sporothrix globosa yeast cells

**Acharya Balkrishna**[1,2,3]**, Sudeep Verma**[1]**, Vallabh Prakash Mulay**[1]**, Ashish Kumar Gupta**[1]**, Swati Haldar**[1]**, Anurag Varshney**[1,2,4]*

**1** Drug Discovery and Development Division, Patanjali Research Institute, Haridwar, India, **2** Department of Allied and Applied Sciences, University of Patanjali, Patanjali Yog Peeth, Haridwar, India, **3** Patanjali Yog Peeth (UK) Trust, Glasgow, United Kingdom, **4** Special Centre for Systems Medicine, Jawaharlal Nehru University, New Delhi, India

* anurag@prft.co.in

**Data Availability Statement:** All relevant data are within the manuscript and its Supporting Information file.

## Abstract

Chronic topical cases of Sporotrichosis, a chronic fungal infection caused by the ubiquitously present cryptic members of the *Sporothrix* species complex, are treated with oral administrations of itraconazole. However, severe pulmonary or disseminated cases require repeated intra-venous doses of amphotericin B or even surgical debridement of the infected tissue. The unavoidable adverse side-effects of the current treatments, besides the growing drug resistance among *Sporothrix* genus, demands exploration of alternative therapeutic options. Medicinal herbs, due to their multi-targeting capacity, are gaining popularity amidst the rising antimicrobial recalcitrance. *Withania somnifera* is a well-known medicinal herb with reported antifungal activities against several pathogenic fungal genera. In this study, the antifungal effect of the whole plant extract of *W. somnifera* (WSWE) has been explored for the first time, against an itraconazole resistant strain of *S. globosa*. WSWE treatment inhibited *S. globosa* yeast form growth in a dose-dependent manner, with IC$_{50}$ of 1.40 mg/ml. Minimum fungicidal concentration (MFC) was found to be 50 mg/ml. Sorbitol protection and ergosterol binding assays, revealed that anti-sporotrichotic effects of WSWE correlated well with the destabilization of the fungal cell wall and cell membrane. This observation was validated through dose-dependent decrease in overall ergosterol contents in WSWE-treated *S. globosa* cells. Compositional analysis of WSWE through high performance liquid chromatography (HPLC) exhibited the presence of several anti-microbial phytochemicals like withanone, withaferin A, withanolides A and B, and withanoside IV and V. Withanone and withaferin A, purified from WSWE, were 10–20 folds more potent against *S. globosa* than WSWE, thus, suggesting to be the major phytocompounds responsible for the observed anti-sporotrichotic activity. In conclusion, this study has demonstrated the anti-sporotrichotic property of the whole plant extract of *W. somnifera* against *S. globosa* that could be further explored for the development of a natural antifungal agent against chronic Sporotrichosis.

**Funding:** This research received no external funding. The work was carried out with internal funds received from Patanjali Research Foundation Trust, Haridwar, India. The external funders had no role in study design, data collection and analysis, decision to publish, or preparation of the manuscript.

**Competing interests:** I have read the journal's policy and the authors of this manuscript have the following competing interests: Acharya Balkrishna is a trustee in Divya Yog Mandir Trust, Haridwar, India that governs Divya Pharmacy, Haridwar. In addition, Acharya Balkrishna is one of the founding promoter and holds an honorary managerial position in Patanjali Ayurved Ltd, Haridwar, India. Divya Pharmacy and Patanjali Ayurveda Ltd commercially manufacture and sell several ayurvedic products, some of which include Withania somnifera as herbal component. Divya Pharmacy and/or Patanjali Ayurved Ltd, Haridwar, India were not involved in any aspect of this study. In addition, Anurag Varshney is an adjunct professor in Department of Allied and Applied Sciences, University of Patanjali, NH-58, Haridwar-249405, Uttarakhand, India; and in Special Centre for Systems Medicine, Jawaharlal Nehru University, New Delhi-110067, India.

## Author summary

Sporotrichosis, commonly known as 'rose gardener's disease' is a rare but chronic fungal infection caused by several pathogenic members of *Sporothrix*. Although rare, Sporotrichosis can spread to the lungs or whole body, and thus, could be fatal. However, due to low frequency of its occurrence, scientific research on Sporotrichosis has been rather limited. To exacerbate the situation, many *Sporothrix* members have developed resistance against the common anti-fungal agents used to treat fungal infections. Many medicinal herbs are being explored for their effectiveness against drug-resistant microorganisms. In this connection, the well-known medicinal plant, *Withania somnifera*, commonly known as Winter cherry in English and Ashwagandha in Hindi, has been explored for its anti-fungal potentials against several pathogenic fungi. However, to the best of our knowledge, anti-fungal effect of *W. somnifera* against *Sporothrix* has not been assessed yet. Therefore, this study was conducted to investigate the anti-fungal potentials of *Withania somnifera* whole plant extract (WSWE) against *Sporothrix globosa*, known to cause Sporotrichosis in Asia. This study demonstrated significant anti-sporotrichotic effect of WSWE, which could be further explored for developing an alternative natural treatment for Sporotrichosis.

## Introduction

Sporotrichosis, colloquially known as "rose gardener's disease" is primarily a chronic cutaneous mycosis caused by fungal pathogens, that belong to a species complex of *Sporothrix schenckii*, consisting of five clinically relevant cryptic species, *S. globosa*, *S. brasiliensis*, *S. luriei*, *S. schenckii sensu stricto* and *S. mexicana* [1,2]. Cutaneous Sporotrichosis is initially observed as a benign sub-cutaneous growth, where the pathogenic propagules (vegetative part or conidia) have entered the skin through small cut, scrape or bite from an infected cat. However, this initial painless manifestation turns into an open sore or ulcer within 2–3 months, with chronic sores appearing around the original one [1]. Although subcutaneous transmission is the primary mode of infection for Sporotrichosis, airborne cases are also reported [3]. These fungal pathogens are ubiquitously present in soil and on plant matters, particularly, rose bushes, sphagnum moss and hay, and can have clinical manifestations in both immunocompromised and immunocompetent individuals [4,5]. Most medically relevant *Sporothrix spp* grow in soil [6]. Pulmonary and disseminated are the other two forms of Sporotrichosis. Pulmonary Sporotrichosis is rare and has cough, chest pain, shortness of breath, and fever as clinical symptoms, whereas, those of disseminated Sporotrichosis depend on the organ being affected [1]. Some of these fungi, namely, *S. brasiliensis* [7], S. schenckii [8] and *S. globosa* [9] are thermo-dimorphic and exist as saprophytic mycelial form at 25–28˚C and the pathogenic yeast form at 36–37˚C, the latter matching the core temperature of their warm-blooded hosts, that include humans and animals [10]. This temperature-induced morphological transformation, particularly, in case of *S. brasiliensis* and *S. globosa* is crucial for establishing virulence [7,9,10]. By increasing the ambient temperature from 25˚C to 37˚C, morphological switching from mycelial to yeast form can be triggered in culture [8].

Sporotrichosis is prevalent mostly in the warm, humid tropical and subtropical regions where the climatic conditions are conducive for saprophytic growth of the pathogen, such as, the sub-Himalayan regions of India [11]. In spite of distinctive genetic differences, the cryptic species of *S. schenckii* are morphologically indistinguishable. However, they have different

ecological preferences, which has led to worldwide distribution of this fungal pathogen, making Sporotrichosis a global health problem. Causative *Sporothrix* species of Sporotrichosis exhibits distinct endemicity, like, Brazil has only *S. brasiliensis*, Australia, Mozambique, United Kingdom, Netherlands, Germany, France, Venezuela, Peru, Bolivia and Argentina have only *S. schenckii* and Portugal only *S. mexicana*. Except for *S. brasiliensis*, all other species also exist outside their endemic areas, like, *S. schenckii* has been found in the USA, Mexico, Colombia, Brazil, South Africa, Italy and Japan. Similarly, outside Portugal, *S. mexicana* is reported in Italy and Mexico. *S. luriei* is endemic only to South Africa. *S. globosa* is endemic to India, China, Spain and Guatemala, besides, being prevalent in Japan. Out of the five cryptic species, the predominant ones are *S. globosa*, *S. brasiliensis* and *S. schenckii*, respectively, dominating the endemicity in Asia and Europe, Brazil (South America) and both Americas. USA, Mexico, Colombia and Italy have also reports of *S. globosa*. Such a global distribution of *Sporothrix* spp., has created an endemic belt of the disease, Sporotrichosis, starting in the USA, covering Latin America, southern part of the African continent all the way to Australia. India, China and Japan, although not connected to this belt, are identified as endemic to Sporotrichosis. In fact, China is recognized as an hyperendemic zone for the disease, just like, Brazil and South Africa [5]. Sporotrichosis is rare with chronicity but there have been sporadic outbreaks in the past century, and the severe cases involving pulmonary and disseminated forms can be potentially fatal, making this disease a serious public health concern [12]. Despite sub-acute infection, the chronicity of Sporotrichosis together with increasing resistance towards available antifungal agents [13], makes this fungal infection a medical challenge. Topical cases of cutaneous Sporotrichosis are more chronic than fatal and are treated with itraconazole whereas, more severe ones require intra-venous administration of amphotericin B. But, this drug is known for its adverse side-effects. Pulmonary Sporotrichosis may also require surgical removal of the infected lung tissue [1]. More than half (56%) of epidemiological distribution of *S. globosa* is in Asia [14]. Therefore, because of the extent of global spread, endemicity to Asia, particularly, India and associated treatment challenges, the cryptic species, *S. globosa* was selected as the target *Sporothrix spp.* for the current study on antifungal efficacy of WSWE. Genetic diversity of the cryptic species dictates their antifungal susceptibility profiles [15]. Amidst this, the members of *Sporothrix* species complex are rapidly developing drug resistance to the currently available antifungal agents [13]. Therefore, there is a pressing need for identifying alternative antifungal agents against *Sporothrix spp*.

The therapeutic application of herbs and their derivatives is an ancient tradition of India and China, and is gradually being accepted in the West [16]. Ferreira *et al* (2019) have explored the anti-fungal effects of α- and β-2,3-dihydrofuranaphthoquinones against *S. brasiliensis* and *S. schenckii* [17]. Some early exploratory studies on leaf extracts of *Terminalia prunioides*, *T. brachystemma*, *T. sericea*, *T. gazensis*, *T. mollis*, *T.sambesiaca* [18] and *Combretum nelsonii* [19] demonstrated anti-fungal effects on *Sporothrix schenckii in vitro*. Acetone extracts of *Combretum imberbe* Wawra, *Combretum nelsonii* Duemmer, *Combretum albopunctatum* Suesseng, and *Terminalia sericea* Burch ex DC along with asiatic and arjunolic acids isolated from *C. nelsonii* exhibited potent *in vivo* fungal infection clearing ability in the wounds of immunocompromised Wistar rats [20]. Preliminary microbiological studies, using agar diffusion, broth microdilution, bioautography and agar cup methods, identified 141 species of plants, mostly from families Combretaceae (30%), Asteraceae (11%) and Lamiaceae (7%), with anti-sporotrichotic potential [16]. However, *Withania somnifera* was not there in the list, despite its reported anti-fungal potentials [21–24]. *W. somnifera* is one of the most extensively used medicinal plants in Ayurveda, the traditional Indian medicinal system [25]. Classical Ayurvedic texts mostly recommend root extracts of *W. somnifera* for the management of various disorders. A number of research articles on antifungal potentials of different parts of *W*.

*somnifera*, against several pathogenic fungi, except *Sporothrix*, are available [22]. Besides, these studies did not investigate the modes-of-actions, the outcomes of which would have been instrumental in addressing the increasing drug resistance within *Sporothrix spp*. Moreover, most of these studies were conducted against *Sporothrix schenckii* [16]. However, with the identification of other pathogenic species from this complex, along with their corresponding endemicity, it becomes quite important to explore therapeutic options against them as well.

The current study was conducted to explore the antifungal potentials of the extract of *W. somnifera* whole plant (WSWE) against an itraconazole-resistant *Sporothrix globosa* strain. The present study was designed to decipher the mode-of-action of the *W. somnifera* whole plant extract (WSWE) against the pathogenic yeast form of *S. globosa*.

## Methods

### Chemicals, fungal media, and fungal strain

The fungal culture medium was procured from Himedia (HiMedia, Mumbai, India). The pathogenic strain of *S. globosa* (NCCPF220119) was obtained from the National Culture Collection of Pathogenic Fungi (NCCPF), Post Graduate Institute of Medical Education & Research (PGIMER), Chandigarh, India. Fungal cells were reactivated from glycerol stock, first by streaking on brain heart infusion (BHI) agar plate for single colony and subsequently, by overnight culturing in BHI broth at 37°C. All *S. globosa* cultures were either grown on BHI agar plates or in BHI broth. Withanoside IV, withanoside V, withaferin A and withanone standards were purchased from Natural Remedies Pvt. Ltd. (Bengaluru, India) whereas, those for withanolides A and B were procured from Sigma-Aldrich (Bengaluru, India). Ergosterol standard was procured from Tokyo Chemical Industry (Tokyo, Japan).

### Procurement of *W. somnifera* plant and preparation of the extract WSWE

Different parts, namely, leaves, stems, fruits, and flowers of *W. somnifera* were collected (Collection No. 3415) from the herbal garden of Patanjali Research Institute, Haridwar, India. Identification and authentication of the plant material was done by taxonomists at Patanjali Research Foundation Herbarium (PRFH) and recorded under the Accession No. 5606. The authenticated plant materials were washed with distilled water and air-dried at room temperature for 2 weeks. The dried parts were then ground into a uniform powder. Subsequently, hydro-methanolic extract was prepared by soaking 100 g of the powdered plant material in hydro-methanolic (80 water: 20 methanol) solvent at room temperature for 48 h. The extract was filtered through Whatmann filter paper No. 42 and concentrated using a rotary evaporator (Heidolph Laborota 4001 Rotary Evaporator System, Marshall Scientific, Hampton, NH, USA) with heating on a water bath at 50°C to obtain the final extract with semi-solid consistency.

### Determination of inhibitory concentration through broth microdilution method

The antifungal strength of WSWE against *S. globosa* was determined from the inhibitory concentration (IC) values obtained through the broth microdilution method, in compliance with the recommendations of the Clinical and Laboratory Standards Institute (CLSI, 2015). 96-well microplates were used to conduct the broth microdilution assay. The test article stock solution was prepared by dissolving 50 mg of WSWE in 1 ml of 1X PBS. Stock solutions were further diluted 2-fold serially, and 100 µl of each dilution was added to designated well containing BHI broth resulting in test concentrations varying from 0.19 to 50 mg/ml. 100µl of the *S. globosa* liquid culture in BHI broth was added to each test well to a final cell density of $10^6$ CFU/

ml. The control wells meant to serve as corresponding blanks for the concentration range of the test article received only BHI broth without any cell. The plates were incubated at 37˚C for 24 hours following which absorbance at 660 nm was measured using a microplate reader (Envision, Perkin Elmer, Waltham, MA, USA). The percentage inhibition of fungal growth by WSWE was determined by using the following equation:

$$\% \text{ growth inhibition} = 100 - [(A_c - A_t)/A_c \times 100]$$

Where $A_c$: Absorbance of the control; $A_t$: Absorbance of test extract.

Percent growth inhibition was represented graphically as a function of increasing log concentrations of the WSWE. Using the inbuilt non-linear regression analysis option of GraphPad Prism software (GraphPad Prism 7.0, San Diego, CA, USA), the $IC_{50}$ and $IC_{80}$ values of WSWE, the concentrations at which the extract inhibited 50 and 80% fungal growth, respectively, were determined. Minimal fungicidal concentrations were determined by plating the cultures of *S. globosa* treated with different concentrations of WSWE on BHI agar plates. The treated cultures were 1000-fold diluted before plating to obtain countable colonies. The dilution factor was taken into account while calculating the colony forming units (CFU)/ml.

Broth microdilutions for control antifungals, itraconazole and amphotericin B were conducted as above and the 2-fold serially diluted test concentrations ranged from 0.25 to 125 μg/ml.

## Time-kill kinetics assay

For time-kill kinetics of WSWE assay, $10^6$ CFU/ml were treated with WSWE and amphotericin B at their respective $IC_{50}$ concentrations at 37˚C. Aliquots of 1.0 ml of the cultures were harvested at different time intervals of 0, 2, 4, 6, 12, 24, 30, 48, 54, 72 and 96 h, and absorbance measured at 660 nm. The control included untreated cells incubated under identical conditions.

## Sorbitol protection assay

Sorbitol protection assay relies on the isotonic protection provided by 0.8 M sorbitol in the medium to the cells whose cell walls have been damaged by the test anti-fungal agent [26]. Due to this protection, in the presence of sorbitol, the efficacy of the anti-fungal agent is apparently reduced as evident from increased $IC_{50}$ value in comparison to that obtained for the same fungus in the absence of sorbitol. For the sorbitol protection assay, *S. globosa* cells were treated for 24 h at 37˚C with different concentrations of either WSWE (2-fold serially diluted range from 0.19 to 50 mg/ml) or amphotericin B (2-fold serially diluted range from 0.25 to 125 μg/ml) in the presence or absence of sorbitol (0.8 M) in the media. Absorbance was measured and inhibitory concentrations determined as described above under the section 'Determination of Inhibitory Concentration through Broth Microdilution method.

## Ergosterol binding assay

Ergosterol binding assay helps to determine whether the test anti-fungal agent affects the fungal cell membrane by sequestering the membrane ergosterol [26]. In this case, the 400 μg/ml ergosterol provided in the culture medium competitively inhibited the test article from binding to the membrane ergosterol, thereby, reducing its anti-fungal efficacy. This is manifested as increased $IC_{50}$ value of the anti-fungal agent in the presence of ergosterol in the medium as compared to that determined in its absence. Ergosterol binding assay was conducted in a similar way as sorbitol protection assay. All parameters were same except that 0.8 M sorbitol in the media was replaced with 400 μg/ml ergosterol.

## Lactophenol cotton blue-trypan blue (LCB-TB) staining and brightfield microscopy

A fungal staining protocol of trypan blue with 0.25% lactophenol cotton blue stain was used. *S. globosa* cells ($10^6$ cells/ml) were treated with $IC_{50}$ concentration of either WSWE or ampB at 37°C for 24 h. The control included cells incubated under identical conditions without any treatment. The cells were harvested post incubation by centrifugation at 3000 rpm for 10 minutes at room temperature, and pellets suspended in 50 μl 1X PBS. To the cell suspensions, 10 μl of stain (0.1% trypan blue with 0.25% lactophenol cotton blue stain) was added and the stained cells were subjected to brightfield microscopy (Axioscope, Carl Zeiss, Germany). As a reference for cell swelling due to cell wall damage and consequent protoplast formation, lyticase treated cells (1 μg/ml of lyticase for $10^6$ cells/ml in a 100 μl reaction for 15 min at 37°C) incubated in distilled water for 30 min were included.

## Compositional analysis of WSWE

Phytochemical composition of WSWE was analyzed through high performance liquid chromatography (HPLC) (Prominence-i LC-2030c 3D Plus, Shimadzu, Japan) using Shimadzu shimpack GIST C18 (5 μm, 4.6*250 mm) column. The test sample was prepared by dissolving 0.5 gm of WSWE in 10 ml of methanol and sonicated for 30 min. The solution was clarified by centrifugation at 10000 rpm for 5 minutes. Prior to injection, each sample was filtered through a 0.45 μm filter. All standards were dissolved in methanol and diluted to 50 μg/ml working concentrations. Chromatographic elution was carried out at 27°C temperature under binary gradient conditions. The flow rate and the injection volume were 1.5 ml/min and 10 μl respectively. The elution was carried out by using a gradient of mobile phases A [0.014% (w/v) $KH_2PO_4$, 0.05% (v/v) $H_3PO_4$ in water] and B (Acetonitrile). Elution started with mobile phases A:B volume percent ratio of 90:10 which changed to 70:30 by 18 min and to 65:35 by 23 min, and was maintained until 30 min. Thereafter, it further changed to 60:40 by 35 min reaching a 50:50 A:B volume percent ratio by 40 min. By 45 and 50 min, this ratio changed to 30:70 and 20: 80, respectively. In next 1 min, it came back to the original A:B percent volume ratio of 90:10 and was maintained for another 4 min, before the next injection. The resultant chromatograph was recorded at 227 nm wavelength.

## Isolation of individual phytocompounds and assessment of anti-sporotrichotic activities

For purifying withanone, the crude WSWE was adsorbed on Si 100–200 mesh at 1:10 ratio between extract and silica to form a slurry. The elution was initiated with chloroform, followed by progressive increase in methanol ratio to proportionately enhance the polarity. Pure withanone was eluted at chloroform: methanol ratio of 99.5:0.5. The fraction containing withanone was then concentrated. Withaferin A was also purified through a similar process in which a dynamic mobile phase of chloroform-hexane was used, beginning with 100% hexane. Concentration of chloroform in the mobile phase was increased in gradients, and withaferin A eluted with 100% chloroform, and was subsequently, concentrated. Purities of isolated withanone and withaferin A was verified through HPLC analyses. Anti-sporotrichotic activities of purified withanone and withaferin A were evaluated through broth microdilution method as described earlier under 'Determination of Inhibitory Concentration through Broth Microdilution method' of this section.

## HPLC analysis of overall ergosterol in *Sporothrix globosa*

Between 10 to 20 mg biomass of untreated *S. globosa* cells and those treated with either amphotericin B (at $IC_{50}$) or different concentrations of WSWE ($IC_{50}$ and 2X $IC_{50}$) was dissolved in 1

ml of methanol by triturating with stainless steel balls and vortexing for 30 min. This solution was then centrifuged for 5 min at 10000 rpm and filtered through 0.45μm nylon filter before injecting into the system. Ergosterol standard (97.5%) was dissolved in methanol to prepare 1000 ppm standard solution. 0.1 ml of this standard solution was diluted to 10 ml to prepare 10 μg/ml working stock. Ergosterol content of the treated and untreated *S. globosa* was analyzed on HPLC (Prominence-i LC-2030c 3D Plus, Shimadzu, Japan). Separation was achieved using a Shodex C18-4E (5 μm, 4.6*250 mm) column subjected to isocratic elution of mobile phase acetonitrile and methanol (80:20) at a flow rate of 1 ml/min. 50 μl of standard and test solution were injected and column temperature was maintained at 25˚C. Wavelength was set 280 nm and sample solution was kept at 4˚C during the analysis.

## Statistical analysis

Data represented as mean ± standard error of mean (SEM) using GraphPad Prism version 7.0 (GraphPad Prism 7.0, San Diego, CA, USA). Statistical significance of the observations was determined using in-built analysis option of the software. Observation was considered to be significant at $p < 0.05$. Experiments were repeated multiple times ($n \geq 2$ to $n \leq 6$) to generate the mean anti-fungal response of WSWE against *S. globosa*.

## Results

### WSWE exhibits antifungal activity against *S. globosa*

Antifungal effect of WSWE was evaluated against the pathogenic yeast form of *S. globosa*. through broth microdilution method. Itraconazole and amphotericin B were included as control antifungal agents. The strain of *S. globosa* used in this study was found to be resistant to Itraconazole (**Fig 1A**). Amphotericin B inhibited 50% growth of *S. globosa* at a concentration of 0.55 μg/ml ($IC_{50}$). At 1.26 μg/ml ($IC_{80}$), amphotericin B impeded 80% of the fungal growth (**Fig 1B**). 1.40 mg/ml ($IC_{50}$) of WSWE was required to suppress 50% of *S. globosa* growth, and for an 80% growth inhibition ($IC_{80}$), this amount was found to be 3.68 mg/ml (**Fig 1C**). When *S. globosa* cells treated with different concentrations of WSWE were plated on non-selective BHI agar plates, the minimum fungicidal concentration (MFC) was found to be 50 mg/ml (**Fig 2A and 2B**). The mechanism of antibiosis can be determined from the ratio of MFC and the minimum inhibitory concentration (MIC) or the concentration of the anti-microbial agent offering visible microbial growth inhibition. So, 99% inhibition ($IC_{99}$) of *S. globosa* growth was calculated through non-linear regression, using the in-built analysis tool of the GraphPad Prism software, and was found to be 34.51 mg/ml. The ratio between MFC and $IC_{99}$ (MIC in this case) being 1.45 ($< 2$), indicates that the mechanism of antibiosis of WSWE is most likely fungicidal [27]. Taken together, these observations exhibited prominent antifungal activity of WSWE against *S. globosa*.

The time-kill curve of the untreated cells shows that the logarithmic growth phase of *S. globosa* spans from 6 to 48 h (**Fig 3A**). Amphotericin B treatment delayed the onset of logarithmic phase by 6 h, so the cells entered exponential growth by 12 h. Furthermore, growth of amphotericin B treated cells started plateauing by 24 h, with very slight increase in growth over next 72 h until 96 h, thereby, reducing the overall duration of exponential phase by almost half when compared to untreated cells. A similar trend was noted for WSWE treated cells whose growth levelled by 24 h, absorbance hovering over 1 to 1.1 until 96 h. However, WSWE treated cells, unlike, their amphotericin B treated counterparts, entered the logarithmic phase at 6 h along with their untreated peers. Regardless, by 24 h, exponential growth of amphotericin B and WSWE treated cells were reduced by ~ 56.4 and 39.9%, respectively, relative to the untreated cells. In the next 24 h, when the untreated cells reached their plateau phase with an

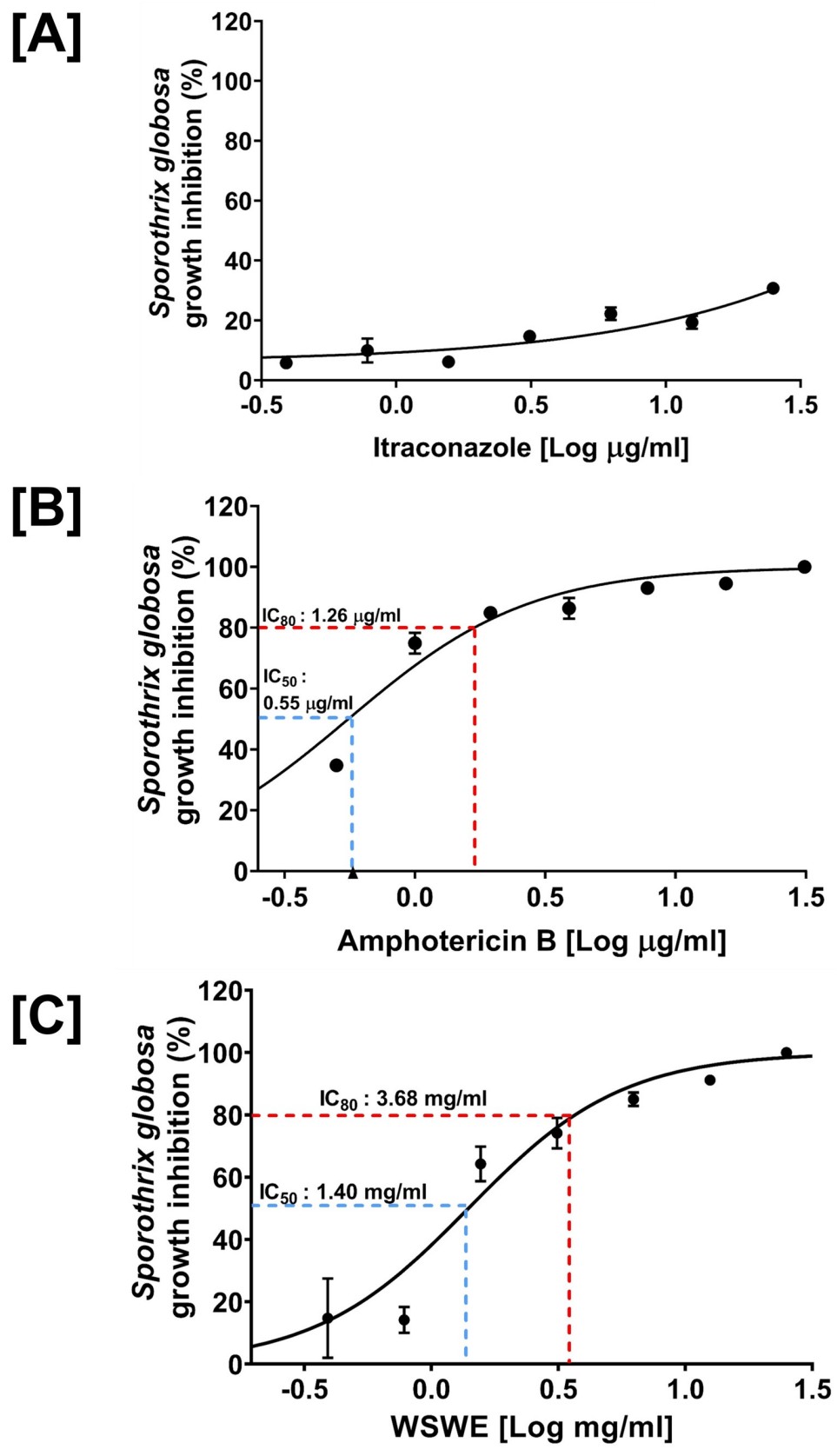

**Fig 1. Antifungal effect of WSWE on *S. globosa*.** [A-C] Dose response curves showing dose-dependent inhibitory effect of itraconazole (A), amphotericin B (B) and WSWE (C) on the growth of *S. globosa* cells. Different inhibitory concentrations, determined through non-linear regression analysis are indicated.

absorbance of 2.1, the reduction in growth of *S. globosa* relative to untreated cells, due to either treatment was more. Amphotericin B and WSWE treated cells showed respectively ~59.1 and 43.9% less growth relative to the control. These observations from the time kill assay indicates the inhibition kinetics of WSWE for *S. globosa* suggesting that it can potentially inhibit colonization of this fungus. Growth rates of the differentially treated and untreated cells were analysed using Malthusian curve fitting over 6, 12 and 24 h. It was observed that for the initial 6 h, untreated and WSWE treated cells almost had similar doubling times of 4.3 and 4.8 h, respectively. However, corroborating with a delayed log phase entry, the amphotericin B treated cells took 8.7 h to double during this initial phase. Over the next 6 h until 12 h, untreated and WSWE treated cells started proliferating faster as evident from reduced doubling times of 2.4 and 2.7 h, respectively. The amphotericin B treated cells still lagged behind with a doubling time of 9.3 h (Fig 3B). By 24 h, while the untreated cells continued to double every 5.8 h, the WSWE treated cells slowed down with a doubling time of 7.1 h. Interestingly, amphotericin B treated cells showed a reduced doubling time of 4.9 h, suggesting an ensuing logarithmic phase (Fig 3B), due to the presence of late growing high tolerant survivors of the drug.

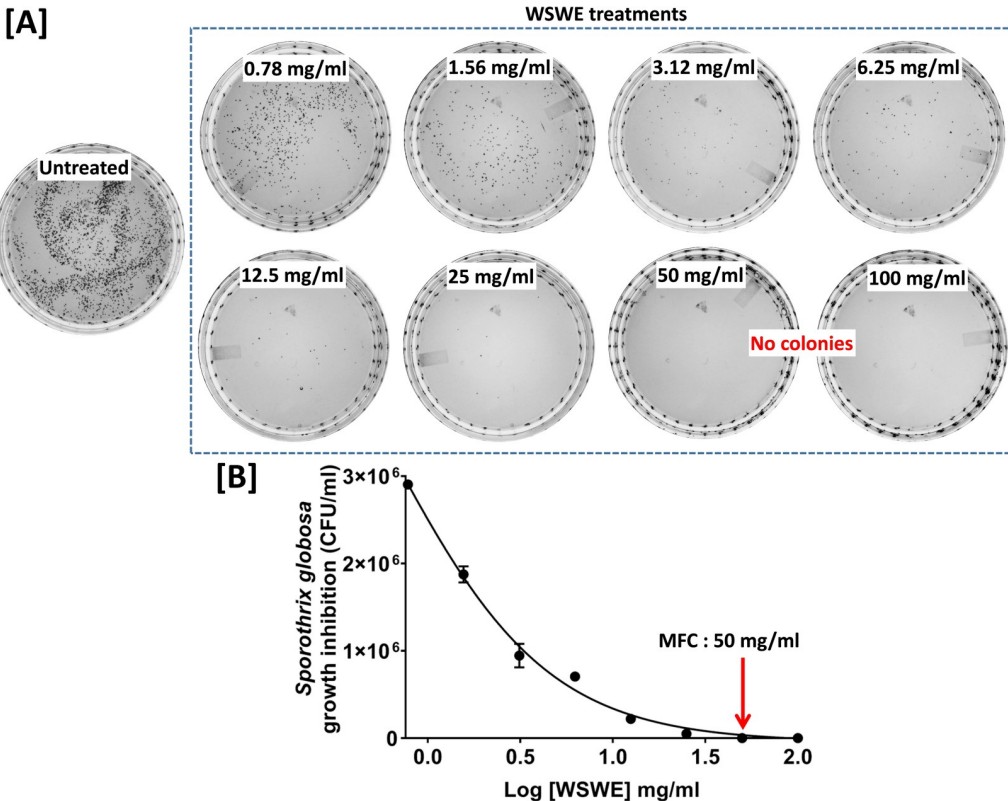

**Fig 2. Determination of minimum fungicidal concentration of WSWE against *S. globosa*.** [A] Representative plate images showing the effect of different concentrations of WSWE on the number of visible colonies of *S. globosa* on agar plates. Absence of visible colonies is indicated. [B] Line graph of CFU/ml *versus* log concentrations of WSWE, quantitatively representing the data shown in (A), with the minimum fungicidal concentration (MFC) indicated with red arrow.

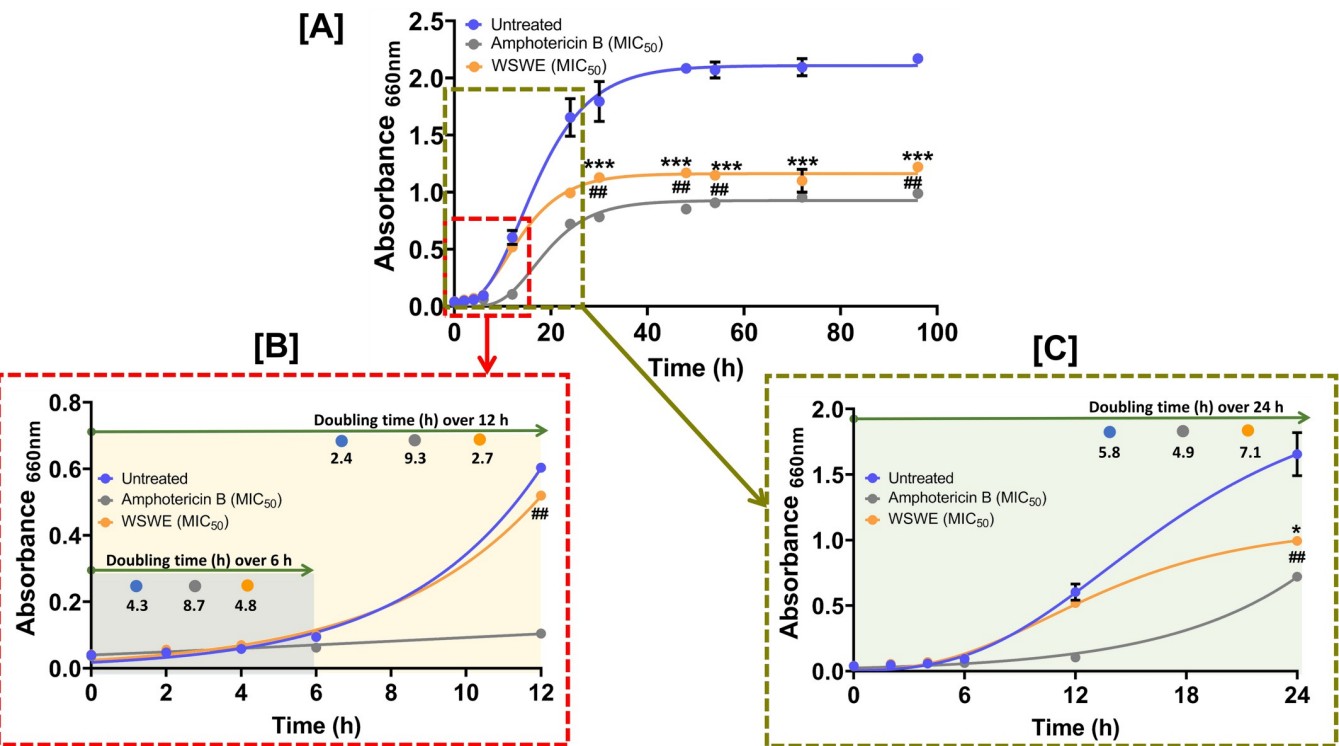

**Fig 3. Time kill kinetics of WSWE against *S. globosa*.** **[A]** Time-kill curve over 96 h growth of untreated and amphotericin B/WSWE treated *S. globosa* cells, obtained through non-linear regression fitting to Gompertz function, showing the inhibitory kinetics of the treatments. **[B, C]** Gompertz function fitted time-kill curves over initial 12 (B) and 24 (C) h of growth, depicting the changes observed in the growth kinetics owing to treatments. The doubling times of the different populations of *S. globosa* cells were determined through Malthusian regression. Each time point represented mean ± SEM of duplicate readings. Statistical significance was determined through 2-way analysis of variance (2-way ANOVA) and represented as * and *** for $p < 0.05$ and 0.001, respectively when comparison was with untreated group, whereas, the comparison with amphotericin B treated group was denoted as ## for $p < 0.01$.

Since, in both the cases of amphotericin B and WSWE, the growth inhibitory effect was observed in actively growing phase of *S. globosa*, it implicated that the observed antifungal activities are most likely owing to interference with pathways responsible for generation of new cells, like duplication of genetic material and/or biosynthesis of cellular structures, such as, cell wall and cell membrane. The *S. globosa* strain being studied has shown resistance to itraconazole, that functions as an antifungal agent by targeting ergosterol biosynthesis [11], whereas, it was susceptible to amphotericin B treatment. In *S. globosa*, like other fungi, amphotericin B is known to bind the membrane-associated ergosterol creating pores leading to leakage of intracellular contents and eventual cell death [11,28]. Given the fact that WSWE affected *S. globosa* cells behaved in a manner similar to amphotericin B, it would be worthwhile to check whether both shares a common mode-of- action as anti-fungal agents. Taking this cue for targeting cell membrane, subsequent experiments were designed to decipher the effect of WSWE on the overall physical protection system of *S. globosa*.

## WSWE targets both cell wall and cell membrane of *S. globosa*

Ergosterol is an important component of *S. globosa* cell membrane, just like other fungi. Likewise, *S. globosa* cells, as in case of other fungi, also enjoy the double protection offered by cell wall [11]. Therefore, sorbitol protection (**Fig 4A**) and ergosterol binding (**Fig 4B**) assays were conducted to evaluate the effect of WSWE treatment on cell wall and cell membrane, respectively [29]. Amphotericin B was included as a reference antifungal control. The rationale

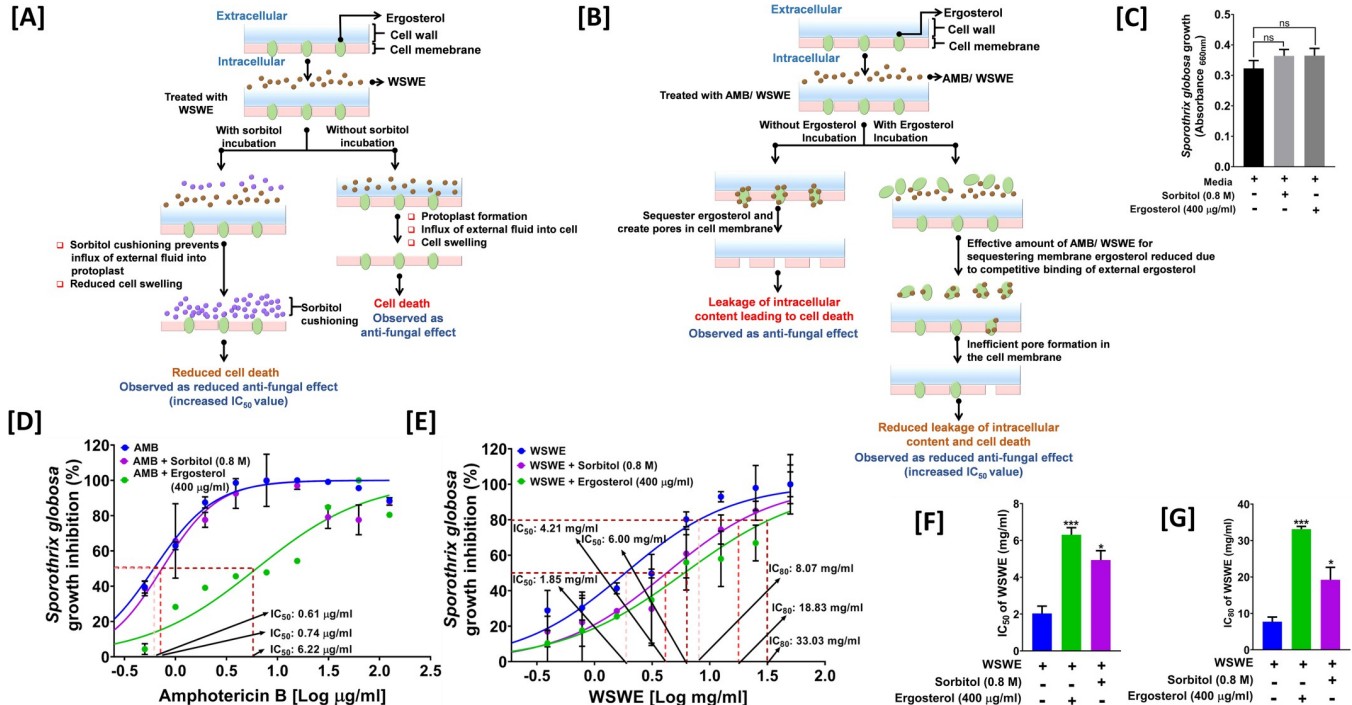

**Fig 4. WSWE targets both cell wall and cell membrane of *S. globosa*. [A, B]** Schematic depiction of the principles behind sorbitol protection (A) and ergosterol binding (B) assays to assess the effect of WSWE treatment on cell wall and cell membrane, respectively, of *S. globosa*. **[C]** Bar graph showing the effect of ergosterol or sorbitol alone on the growth of *S. globosa* cells. **[D, E]** Comparative dose response curves demonstrating the effect of sorbitol and ergosterol on the antifungal efficacies of amphotericin B (D) and WSWE (E) against *S. globosa*. **[F, G]** Bar graphs showing the changes in $IC_{50}$ (F) and $IC_{80}$ (G) values mentioned in (E). Data was individually fitted through non-linear regression and the obtained $IC_{50}$ (F) and $IC_{80}$ (G) values were averaged for plotting. Statistical significance was determined through 1-way ANOVA and represented as * and *** for $p < 0.05$ and 0.001, respectively when compared to treatment with WSWE in the absence of ergosterol and sorbitol.

behind sorbitol protection assay has been depicted in **Fig 4A**. If WSWE disintegrates the cell wall **(indicated through the right arm in Fig 4A)**, it will be essentially generating protoplasts, for which the normal media will act as a hypotonic solution resulting in cell swelling and finally, bursting, However, if sorbitol is provided in the extracellular milieu **(left arm of Fig 4A)**, a cushioning effect ensues as soon as cells form protoplast upon losing cell wall due to WSWE treatment. This will prevent the protoplasts from swelling and bursting. Experimentally, the second scenario will be manifested as reduced fungicidal effect when compared to the first. There will be an increase in the $IC_{50}$ value of WSWE in the presence of sorbitol. Like the sorbitol assay, the ergosterol binding assay also relies upon the shift in $IC_{50}$ value to assess the effect on cell membrane **(Fig 4B)**. In this case, a fungicide, like, amphotericin B which binds the membrane-associated ergosterol creates pores in the cell membrane resulting in intra-cellular leakage and eventual cell death **(left arm of Fig 4B)**. But, if ergosterol is supplied in the extracellular milieu, through competitive binding the extracellular population of ergosterol will significantly reduce the fungicide available for binding the membrane-associated ergosterol. Consequently, this will lead to inefficient pore formation in the fungal cell membrane and manifest as reduced antifungal effect (increased $IC_{50}$ value) of the fungicide **(right arm of Fig 4B)**. Sorbitol or ergosterol alone did not affect the growth of *S. globosa* cells **(Fig 3C)**. A mere 1.21-fold increase in $IC_{50}$ value of amphotericin B observed in the presence of sorbitol, confirmed that it did not affect the cell wall to generate protoplast **(Fig 4D)**. However, a 2.28-fold increase in $IC_{50}$ value of WSWE in the presence of sorbitol, indicated that WSWE

treatment resulted in cell wall disintegration and protoplasting (**Fig 4E**). A similar increase (2.33 fold) was noted for $IC_{80}$ concentration of WSWE in the presence of sorbitol (**Fig 4F**). The observed increase in the $IC_{50}$ values of WSWE in the presence of ergosterol and sorbitol was statistically significant. In case of amphotericin B (known to act through binding membrane-associated ergosterol), $IC_{50}$ value increased by 10.2-fold in the presence of ergosterol in the extracellular milieu (**Fig 4D**). WSWE also showed a decrease in its antifungal activity in the presence of ergosterol in the medium as evident from 3.24 and 4.09-fold increases in its $IC_{50}$ (**Fig 4F**) and $IC_{80}$ (**Fig 4G**) values. Nevertheless, the increases in the $IC_{80}$ values of WSWE in the presence of sorbitol or ergosterol are statistically significant. Furthermore, in order to verify these observations, effect on overall content of ergosterol in *S. globosa* cells treated with amphotericin B or different concentrations of WSWE was analysed through HPLC (**Fig 5A**). Just like, amphotericin B treatment reduced the overall content of ergosterol in *S. globosa* cells, so did, WSWE, in a dose-dependent manner (**Fig 5B**). Altogether, these observations demonstrated that while amphotericin B affected only cell membrane, WSWE affected both cell wall and cell membrane of *S. globosa*.

For qualitative validation of these observations, microscopic analysis of *S. globosa* cells double stained with trypan blue and lactophenol cotton blue was conducted. Trypan blue does not stain the cytosol of live cells, but can stain the chitin and glucans present in fungal cell wall. When trypan blue is used in conjunction with lactophenol cotton blue, since, phenol present in the latter kills the cells, they become permeable to trypan blue, as evident from the untreated cells (**Fig 6A**) [30]. The acetic acid present in lactophenol cotton blue preserves the fungal structures while the cotton blue dye stains the chitin in the cell wall. Therefore, normal untreated *S. globosa* cells when double stained with trypan blue and lactophenol cotton blue, appeared blue with greenish outlines (**Fig 6A**). This set up is very appropriate for analysing cell swelling in response to external hypotonicity (**Fig 6B**). Lyticase treatment enzymatically destroys cell wall glucans due to its endoglucanase and protease activities [31]. Therefore, lyticase treated *S. globosa* cells when incubated in hypotonic solution swell. The trypan blue stained cytosolic content appears as blue patches separated by clearly visible unstained zone from the cell membrane (**Fig 6B**). This validated the suitability of our experimental set-up for assessing alterations in fungal cell wall and cell membrane. Trypan blue stained cytosolic contents of amphotericin B treated cells were found to have leaked out of the cells (**Fig 6C**). In case of WSWE treatment, in addition to leakage of intracellular contents, the cells were found to be swollen (**Fig 6D**). Hence, these microscopic analyses confirmed our observations from sorbitol protection and ergosterol binding assays.

## Compositional analysis of WSWE

It is indeed intriguing to observe that WSWE is effective against both cell wall and cell membrane. To the best of our knowledge, no reported antifungal agent is known to have this dual effect. This prompted us to evaluate the chemical composition of WSWE. HPLC analysis showed that WSWE contained 0.463% (w/w) withanone, followed by almost equal amounts of withanoside V (0.096% w/w) and IV (0.095% w/w), 0.049% w/w of withaferin A and withanolide A and traces of withanolide B (0.001% w/w) (**Fig 7**). Two of these phytocompounds, withanone (**Fig 8A**) and withaferin A (**Fig 8B**), were purified from WSWE and were tested for their anti-sporotrichotic effects. The $IC_{50}$ values for withanone and withaferin A against *S. globosa* were found to be 130 and 94 μg/ml, respectively (**Fig 8C and 8D**), indicating >10 fold higher anti-sporotrichotic effect compared to WSWE. Even the observed $IC_{80}$ values of withanone and withaferin A (287.08 and 160.06 μg/ml, respectively) were in micrograms as compared milligrams in case of WSWE. This clearly indicated that withanone and withaferin A could be responsible for the observed anti-sporotrichotic effect of WSWE.

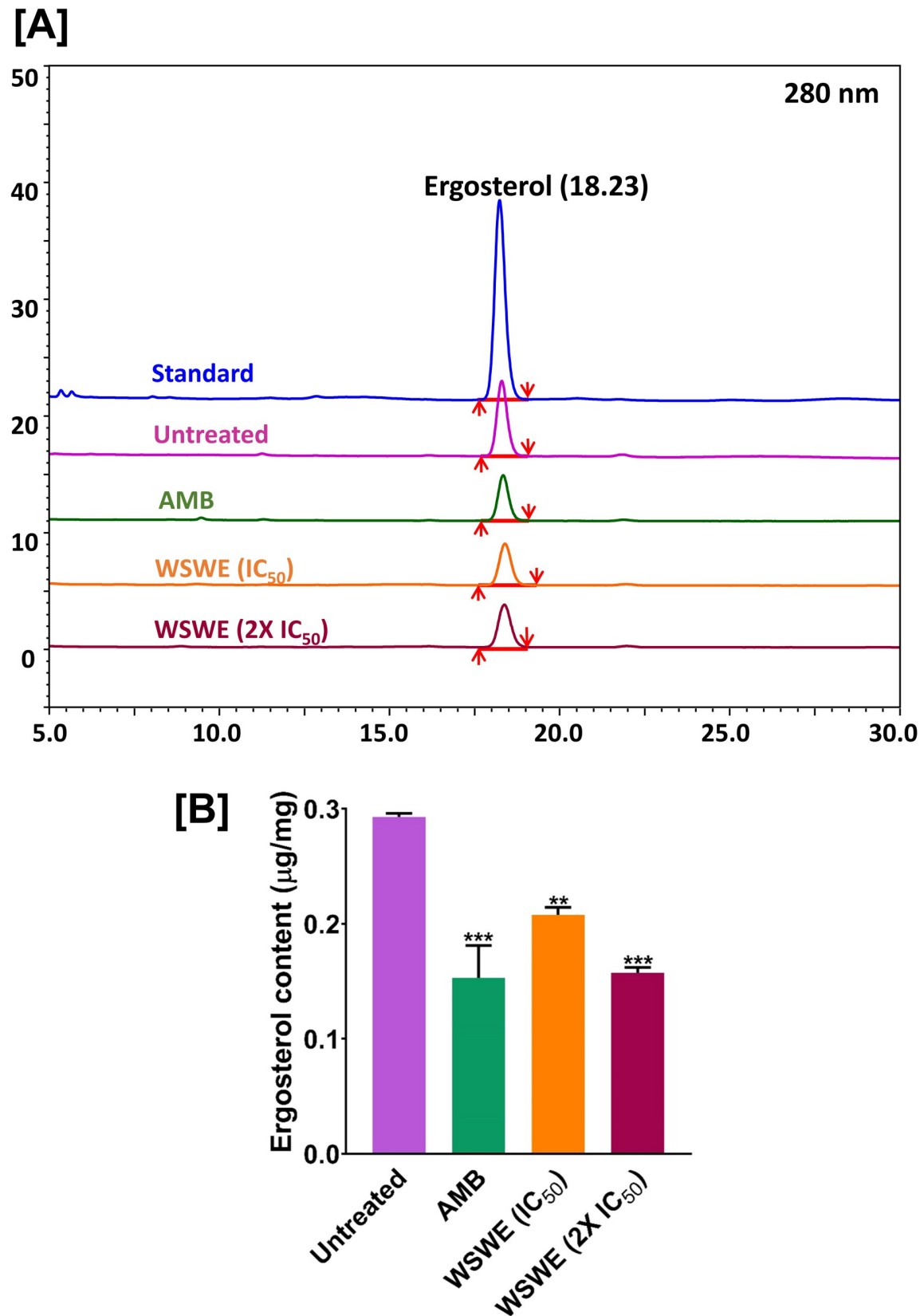

**Fig 5. WSWE treatment reduced total ergosterol amount in *S. globosa* cells.** [A] Mirror HPLC chromatograms recorded at 280 nm, aligned with standard (blue), showing the ergosterol peaks in untreated *S. globosa* cells (purple) and those differentially treated with amphotericin B (AMB) (green) or varying concentrations of WSWE [orange ($IC_{50}$) and maroon ($2X\ IC_{50}$)]. [B] Bar graph showing the decrease in the concentration of ergosterol upon AMB or WSWE treatment.

## Discussion

A decade and a half ago, antifungal resistance was being recognized in the context of different species of the genus *Candida*, but, now other fungal genera have entered the list, *Aspergillus* and *Sporothrix* being the most prominent ones [32,33]. While on a positive note, recent studies report that *S. brasiliensis* is responsive towards amphotericin B and terbinafine [34] and less likely to develop resistance to azole group of fungicides [35], unfortunately, same cannot be claimed for the other species, particularly, *S. globosa* as evident from **Fig 1A**, that shows its resistance to itraconazole. *S. brasiliensis* has not spread globally whereas, *S. globosa* has been reported across Asia, Africa and both the Americas [5]. Mechanism of antimicrobial resistance in different fungi is conserved [33], and can be acquired through molecular alterations with underlying common principles. Altered drug affinity and target abundance, super-active efflux pumps leading to reduced intracellular drug levels and biofilm formation are the main mechanisms for acquiring resistance to antifungal agents [33,36]. The gravity of medical challenge posed by the rapid rise in the antimicrobial resistance of *S. globosa* has prompted scientists to explore complex antibody mediated therapeutic options [37]; the feasibility of the technique in

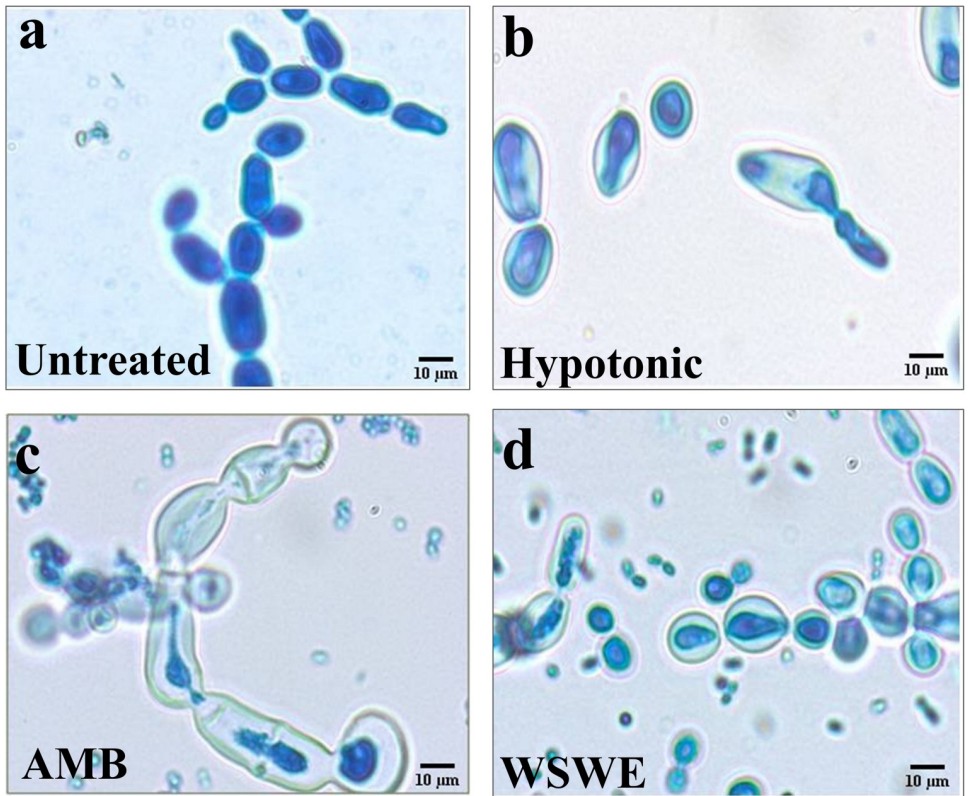

**Fig 6. Micromorphological impact of WSWE treatment on *S. globosa* cells.** Micrographs showing trypan blue lactophenol cotton blue stained untreated (a), hypotonically treated protoplasted (b), amphotericin B (AMB) treated (c) and WSWE treated (d) *S. globosa* cells.

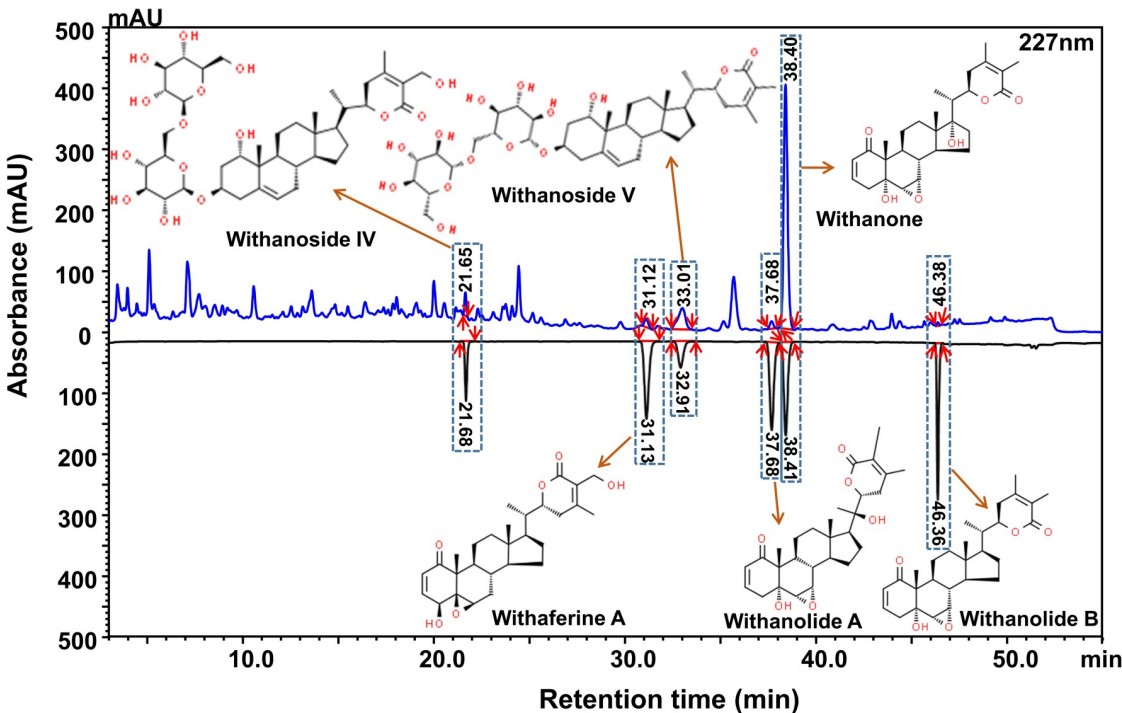

**Fig 7. Phytochemical compositional analysis of WSWE.** Overlap HPLC chromatograms of standard mix (black) and WSWE (blue) recorded at 227 nm. Identified phytocompound peaks are indicated with orange arrows pointing towards respective chemical structures.

question being subject to meticulous evaluation and rigorous validation. As an alternative, antifungal potentials of phytochemicals are also being explored [38,39]. Being multi-targeting in nature, phytochemicals are less prone to obsolescence due to rising anti-microbial resistance [40,41]. It is easier for microbes to evade single targets as are the cases with conventional anti-microbial agents. It becomes challenging for the microbes to develop recalcitrance towards phytochemicals with multiple targets [40]. In this regard, extracts of *Withania somnifera* has been tested for antifungal property, against several pathogenic fungi, but those from genus *Sporothrix* [22,42].

Although, a couple of reports mention about the antifungal activity of root and leaf extracts of *W. somnifera*, available knowledge regarding the precise phytochemical being responsible for such an activity is limited. WSWE, used in the current study, contained withanone, witha-noside IV and V, withaferin A, withanolide A and B, all of which are polyphenols. Flavonoids, a group of polyphenols are known to act as antifungal agents by inhibiting efflux pumps, cell division, nucleic acid and protein synthesis, cell wall formation, disrupting cell membrane and inducing mitochondrial dysfunction [43]. As expected, isolated withanone and withaferin A were found to be more efficacious as anti-sporotrichotic agents compared to the WSWE source. For an insight into the mode-of-action, effect of WSWE treatment on cell wall and cell membrane of *S. globosa* was checked through sorbitol protection and ergosterol binding assays, respectively. Reduced antifungal efficacy in both the cases reflected that WSWE affected both cell wall and cell membrane. Our experimental set-up helps us to understand that WSWE treatment induced cell wall deformation and increased cell permeability (**Fig 9**). In fact, the observed reduction in overall ergosterol content of WSWE treated *S. globosa* cells confirms that cell membrane integrity is the target. It is also possible that the phytochemicals present in

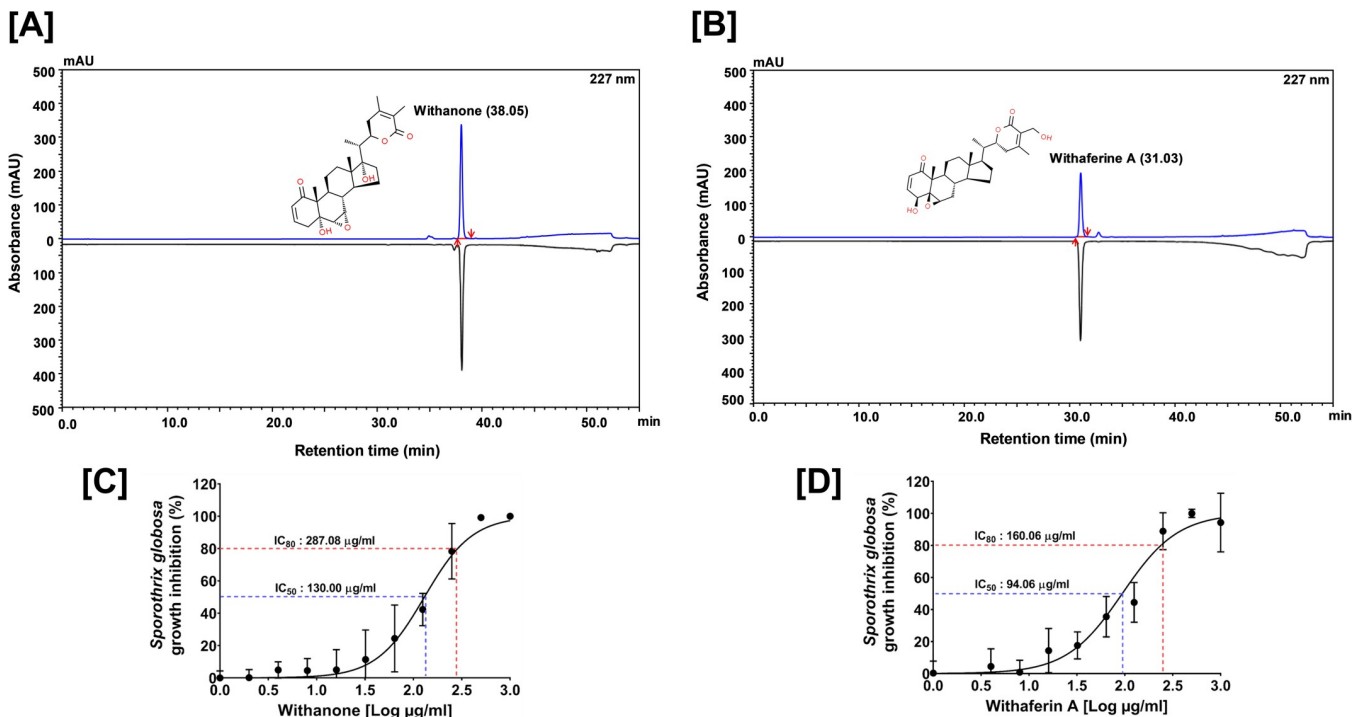

**Fig 8. Effect of withanone and withaferin A purified from WSWE on *S. globosa* cells. [A, B]** Mirror HPLC chromatograms of standard mix (black) and purified withanone (blue) (A)/withaferin A (B) recorded at 227 nm. **[C, D]** Dose response curves of withanone (A) and withaferin A (B) demonstrating anti-sporotrichotic effects against *S. globosa* cells. $IC_{50}$ and $IC_{80}$ values are indicated.

WSWE effectively inhibits the synthesis of glucan and chitin, the much-required building moieties of cell wall. Withaferin A has been shown to inhibit chitin synthesis in insect larva [44]. Chitin is absent in plants, but present in exoskeleton of insects and cell wall of fungi. Although, other mechanisms on antifungal activity are not assessed in this study, nevertheless, being rich in phytochemicals, multiple modes-of-action of WSWE will not be surprising.

The GraphPad Prism software-mediated non-linear regression analyses for obtaining dose response curves for the anti-sporotrichotic effects of amphotericin B and WSWE generated Hill co-efficient values. Positive Hill co-efficients of 2.77 ± 0.76 and 1.35 ± 0.26 for amphotericin B and WSWE, respectively, indicate positive cooperativity of ligand binding in both the cases. The slightly steeper Hill slope of amphotericin B compared to WSWE (2.77 versus 1.35) suggested more cooperativity in case of ligand binding of amphotericin B which might eventually translate into improved drug response. In hindsight, higher cooperativity for ligand binding suggests increased target specificity with increased chances of drug recalcitrance. Lower target specificity as shown by less steep Hill slope/lower Hill co-efficient in case of WSWE might suggest broader target specificity and thereby, lesser chances of evolving drug resistance [40]. Additionally, this creates a possibility for using WSWE in combination with amphotericin B in treating Sporotrichosis, wherein, the latter can effectively eliminate the fungus growing slowly under the effect of the former.

This study has offered several important leads for future studies to develop WSWE into a phytotherapeutic for treating Sporotrichosis. For example, the effect of WSWE on host-pathogen interactions, in terms of pathogen infectivity, host immune-modulation to confront infection, etc. could be explored using *in vitro* experimental models of infection. Development of the WSWE containing formulation is yet another avenue opened by this study. Moreover, in

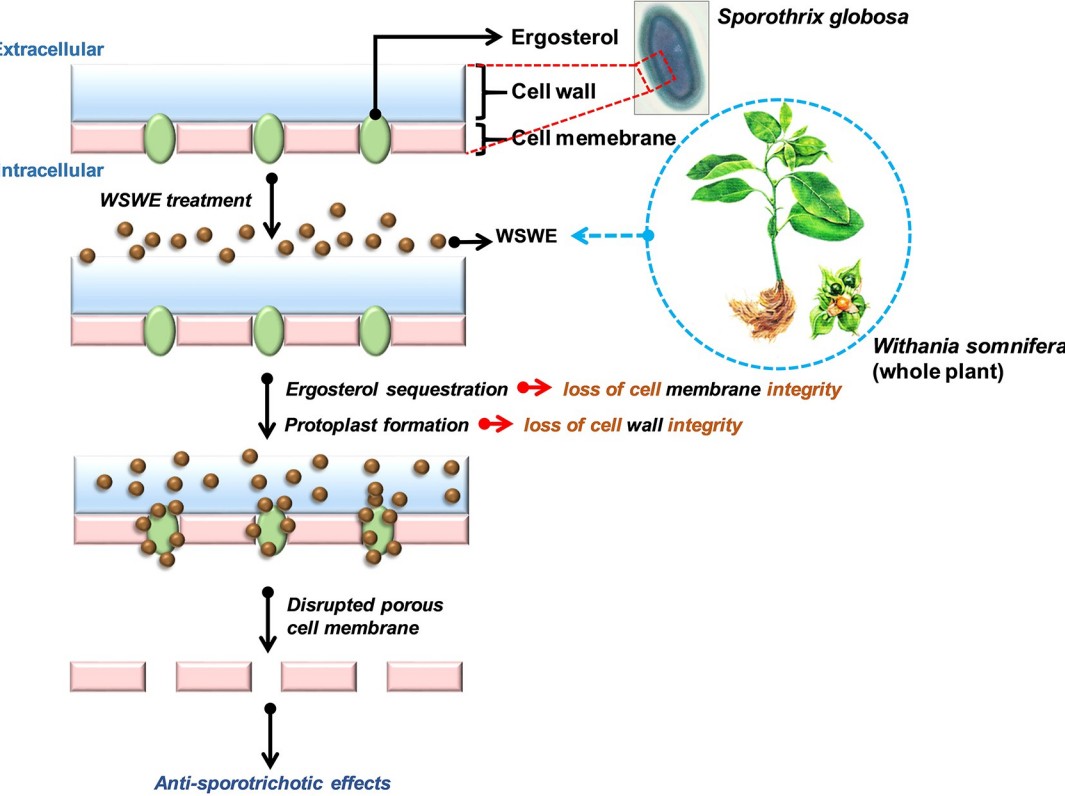

**Fig 9. Proposed model of mode of action of WSWE as anti-fungal agent.** A model depicting possible mode-of-action of WSWE as an anti-fungal agent. *S. globosa* cell wall and cell membrane disintegration is implicated in WSWE treatment, respectively, through protoplasting and ergosterol sequestration. Consequential pore formation and loss of intracellular content are speculated.

detail pre-clinical, toxicological and eventual clinical studies are warranted by the outcomes of this one. Although, all these studies fall beyond the scope of the current manuscript, nevertheless, they do give a limited feel to this chronicle. Besides, the observations reported in this manuscript provide strong rationale for a comprehensive study on the effect of WSWE on multiple clinically isolated strains of *Sporothrix* that would more closely resemble the pathogenic species complex responsible for causing Sporotrichosis.

Sporotrichosis, although rare but quite difficult to treat due to its chronicity. Together with this, the rapidly increasing drug resistance among the pathogenic species necessitates an ardent search for alternative medicines. The available choice of anti-fungal includes amphotericin B, which is known to have associated side effects. This study, even though preliminary, has offered avenues for detailed future studies towards developing WSWE into an anti-sporotrichotic therapeutic. These early findings reported here would be helpful in unconventional treatment strategies with or without currently used anti-fungal agents in combination. This would indeed make a significant impact on translational medicine for treatment and management of Sporotrichosis, and associated public health issues.

In conclusion, current observations suggest that WSWE might be effective against the other members of the pathogenic species complex to which *S. globosa* belongs, but that possibility requires independent evaluations. Taken together, this study has demonstrated the antifungal potentials of whole plant extract of *Withania somnifera* against the drug (itraconazole) resistant pathogenic fungus, *Sporothrix globosa*, that causes chronic cutaneous/sub-cutaneous

infection, Sporotrichosis. These findings suggest that *W. somnifera* extract has potentials of being developed into a remedy for the chronic Sporotrichosis.

## Supporting information

**S1 Data. Excel spreadsheet containing, in separate sheets (for each figure), the underlying numerical datas for Figure panels 1A, 1B, 1C, 2B, 3A, 3B, 3C, 4C, 4D, 4E, 4F, 4G, 5B, 8C and 8D.**
(XLSX)

## Acknowledgments

We would like to thank Dr. Bhaskar Joshi of Patanjali Research Foundation Herbarium, Haridwar, for plant identification and plant material authentication. We thank Dr. Jyotish Srivastava, Ms. Meenu Tomer, Ms. Kanchan Singh, Dr. Sohan Sengupta and Dr. Bhawana Kharayat for their assistance with experiments. We extend our gratitude to Mr. Tarun Rajput, Mr. Gagan Kumar and Mr. Lalit Mohan for their swift administrative supports.

## Author Contributions

**Conceptualization:** Acharya Balkrishna, Anurag Varshney.

**Data curation:** Swati Haldar.

**Formal analysis:** Sudeep Verma, Ashish Kumar Gupta, Swati Haldar.

**Funding acquisition:** Acharya Balkrishna, Anurag Varshney.

**Investigation:** Sudeep Verma, Vallabh Prakash Mulay, Ashish Kumar Gupta, Swati Haldar.

**Methodology:** Sudeep Verma, Vallabh Prakash Mulay, Ashish Kumar Gupta.

**Project administration:** Anurag Varshney.

**Resources:** Acharya Balkrishna, Anurag Varshney.

**Software:** Swati Haldar.

**Supervision:** Swati Haldar.

**Validation:** Swati Haldar.

**Visualization:** Sudeep Verma, Swati Haldar.

**Writing – original draft:** Swati Haldar.

**Writing – review & editing:** Acharya Balkrishna, Swati Haldar, Anurag Varshney.

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
