## [Decision Letter · Decision Letter 0]

8 Feb 2022

Dear Dr. Anurag Varshney,

Thank you very much for submitting your manuscript "Anti-Sporotrichotic Effects of Withania somnifera (L.) Dunal Whole-plant Extracts due to Peripheral Integrity Destabilization of Sporothrix globosa Yeast Cells" for consideration at PLOS Neglected Tropical Diseases. As with all papers reviewed by the journal, your manuscript was reviewed by members of the editorial board and by several independent reviewers. In light of the reviews (below this email), we would like to invite the resubmission of a significantly-revised version that takes into account the reviewers' comments. 

We cannot make any decision about publication until we have seen the revised manuscript and your response to the reviewers' comments. Your revised manuscript is also likely to be sent to reviewers for further evaluation.

Sincerely,

Angel Gonzalez, Ph.D.

Associate Editor

Christine Petersen

Deputy Editor

Reviewer's Responses to Questions

**Key Review Criteria Required for Acceptance?**

**Methods**

-Are the objectives of the study clearly articulated with a clear testable hypothesis stated?

-Is the study design appropriate to address the stated objectives?

-Is the population clearly described and appropriate for the hypothesis being tested?

-Is the sample size sufficient to ensure adequate power to address the hypothesis being tested?

-Were correct statistical analysis used to support conclusions?

-Are there concerns about ethical or regulatory requirements being met?

Reviewer #1: 1. The objectives of the study was clearly articulated with a clear testable hypothesis.

2. The study design was appropriate.

3. As in vitro study so no need of population definition.

4. Statistical analysis was supportive.

Reviewer #2: Include studies on the antisporotrichosis effects of other plants.

Review the study "Ramírez-Soto MC, Aguilar-Ancori EG, Tirado-Sánchez A, Bonifaz A. Ecological Determinants of Sporotrichosis Etiological Agents. J Fungi (Basel). 2018;4(3):95."

The objectives of the study are clearly articulated with a clear testable hypothesis stated.

The study design is appropriate to address the objectives.

There was not statistical analysis. What statistical tests did you use for comparisons?

Reviewer #3: (No Response)

Reviewer #4: Questions in Methods sections: 1. Is fungal culture medium the same than BHI? or are you referring to a different medium? please specify and comment the composition of the medium. 2) The rational behind ergosterol and sorbitol assays is not clear. Please specify. 3) I think the terms MIC50 and MIC80 are wrongly used here. 4) what were the standards used for the HPLC analysis and how is that you choose them? This section is not clear. how you identified withaferin, withanoline etc?

**Results**

-Does the analysis presented match the analysis plan?

-Are the results clearly and completely presented?

-Are the figures (Tables, Images) of sufficient quality for clarity?

Reviewer #1: 1. Analysis presented was match with the analysis plan.

2. Results were clearly and completely presented.

3. Figures and graphs were sufficient and clearly defined.

Reviewer #2: Results is clearly presented. They suggested making comparisons AmB vs. WSWE, and WSWE vs. Untreated, using a statistical test (Figure 2).

Trials are adequate, but I suggest making statistical comparisons (Figure 3).

Reviewer #3: (No Response)

Reviewer #4: Results comply with these points

**Conclusions**

-Are the conclusions supported by the data presented?

-Are the limitations of analysis clearly described?

-Do the authors discuss how these data can be helpful to advance our understanding of the topic under study?

-Is public health relevance addressed?

Reviewer #1: 1. Yes conclusions were supported by data.

2. The author did not mention the limitation of study analysis.

3. The discussion part was weak in supporting the study findings for translational medicine.

4. The study did not addressed the public health importance in major concern.

Reviewer #2: The study should focus on the in vitro test. The first paragraph is out of context., I suggest reviewing.

I suggest expanding the discussion. Include articles on the antisporotrichotic effects of other plants.

Consider to describe that the antisporotrichotic effect of WSWE probably results from the combination of the phytochemical components or from one particular component as well.

They should also discuss further testing to identify the main phytochemical component of WSWE with antisporotrichotic activity.

Describe and discuss the main limitations of your study.

Reviewer #3: (No Response)

Reviewer #4: Authors didnt describe the limitation of their study.

**Editorial and Data Presentation Modifications?**

Reviewer #1: (No Response)

Reviewer #2: Major revision

Reviewer #3: (No Response)

Reviewer #4: I would suggest the authors that

1. In your introduction, tell less about sporotrichosis and more about the public health problem and above all more about the how is currently used? what type of previous studies have been done to study it? what is its range of activity? what previous studies have been published. 

2.Discussion is repetitive compared to the introduction. 

3. The authors didn't explain about how the data analysis was done? how many times these experiments were carried out? 

4. Authors used only one strain, hence they should been using the terms MIC50, MIC90. 

5. I think assays of minimal fungicidal concentration would enrich this WSWE manuscript. As well as TEM 

6. Testing more strains would be useful

**Summary and General Comments**

Reviewer #1: 1.The title was long and confusing, should be revised.

2. Limitation of the study should be discussed.

3. The result should be clearly correlate with future role in disease management(Translational medicine)

Reviewer #2: (No Response)

Reviewer #3: In this study Balkrishna et al., describes the antifungal activity of Withania somnifera whole plant extract against Sporothrix globose and its plausible use in treating Sporotricosis. Although this is a well written manuscript, the study described in this manuscript is limited to the preliminary observation on the crude extract’s antifungal activity against S. globose. There are several reports describing the antifungal activities of W. somnifera extracts. The current study only adds another fungal species to the existing list of fungal pathogens that are susceptible to W. somnifera extracts. Therefore, I am of the opinion that the manuscript in its current form is not suitable for the publication in PLOS Neglected Tropical Diseases and the manuscript need a major revision with potential attempts to identify the active antifungal component of the extract and its plausible mode of action. 

Major Comments: 

1) One of the major weakness of the manuscript is the activities described are limited to the crude extract. The HPLC chromatogram given on the Fig 5, clearly indicates many separable peaks. The authors need to purify these compounds and test the activity of purified peaks to find the potential antifungal molecule/s. Some of the molecules are commercially available, so authors may use commercially available compounds or the purified one. If no individual peaks show any activity against S. globose, the authors could then employ synergistic assays to validate the compounds activity in combinations. If the active peak represents an unknown molecule, the authors are recommended to do the characterization of the same. 

2) On page 13 lines 273-281 and Fig 3 D: The authors used Sorbitol and ergosterol to assess the potential mode of action of WSWE. The results shown in the Figure and text given indicates that there is only a 2-3 folds difference in the MFC50 in the presence of these molecules. The authors are suggested to do statistical analysis of their data given on Fig 3D to validate whether that difference is significant. Also, data given on the Fig3D further indicate that at MFC80 the difference will be meager, unlike AMB data, where difference between +/- Ergosterol is significantly different. I would also suggest the authors to include the growth curves of the S. globose with and without ergosterol and sorbitol as supplementary document, so that readers are aware that these compounds doesn’t affect the overall growth curve. 

Minor comments:

1) Page 12 lines 247-249: “In S. globosa, like other fungi, amphotercin B is known to bind the membrane-associated ergosterol creating pores leading to leakage of intracellular contents and eventual cell death [11,17]. This indicated that WSWE could be affecting cell membrane”. The text or data preceding this statement doesn’t indicate WSWE affect the membrane integrity. Being more active against metabolically active cells doesn’t imply that mode of action of AMB and WEWE are similar. In fact, many known antimicrobial agents show potent activity towards metabolically active cells.

2) Page 17 lines 366-373: It is not clear what the authors are referring to. None of the figures or texts given in the manuscript indicates the given values. Further, the authors need to use caution while claiming a multi target mechanism for WESE extract, as there is no solid evidence for any target.

Reviewer #4: This is as study evaluating the anti fungal effect of whole extract of W. somnifera on a single S. globosa strain. 

Questions in Methods sections: 1. Is fungal culture medium the same than BHI? or are you referring to a different medium? please specify and comment the composition of the medium. 2) The rational behind ergosterol and sorbitol assays is not clear. Please specify. 3) I think the terms MIC50 and MIC80 are wrongly used here. 4) what were the standards used for the HPLC analysis and how is that you choose them? This section is not clear. how you identified withaferin, withanoline etc? 

I would suggest the authors that

1. In your introduction, tell less about sporotrichosis and more about the public health problem and above all more about the how is currently used? what type of previous studies have been done to study it? what is its range of activity? what previous studies have been published. 

2.Discussion is repetitive compared to the introduction. 

3. The authors didn't explain about how the data analysis was done? how many times these experiments were carried out? 

4. Authors used only one strain, hence they should been using the terms MIC50, MIC90. 

5. I think assays of minimal fungicidal concentration would enrich this WSWE manuscript. As well as TEM before and after the efect of WSWE 

6. Testing more strains would be useful 

7. Determination of which component out of the ones found by HPLC is the one with the real anti fungal capacity.

PLOS authors have the option to publish the peer review history of their article (what does this mean?). If published, this will include your full peer review and any attached files.

Reviewer #1: Yes: Dr. Sunil Kumar Gupta

Reviewer #2: No

Reviewer #3: No

Reviewer #4: No
---

## [Editor Report · Decision Letter 1]

9 May 2022

Dear Dr. Varshney

We are pleased to inform you that your manuscript 'Withania somnifera (L.) Dunal Whole-plant Extracts Exhibited Anti-Sporotrichotic Effects by Destabilizing Peripheral Integrity of Sporothrix globosa Yeast Cells' has been provisionally accepted for publication in PLOS Neglected Tropical Diseases.

Best regards,

Angel Gonzalez, Ph.D.

Associate Editor

Christine Petersen

Deputy Editor

---

## [Editor Report · Acceptance letter]

2 Jun 2022

Dear Dr Varshney,

We are delighted to inform you that your manuscript, "*Withania somnifera* (L.) Dunal Whole-plant Extracts Exhibited Anti-Sporotrichotic Effects by Destabilizing Peripheral Integrity of *Sporothrix globosa* Yeast Cells," has been formally accepted for publication in PLOS Neglected Tropical Diseases.

Best regards,

Shaden Kamhawi

co-Editor-in-Chief

Paul Brindley

co-Editor-in-Chief
